# Oxidative Stability of Novel Peptides (Linusorbs) in Flaxseed Meal-Fortified Gluten-Free Bread

**DOI:** 10.3390/foods14030439

**Published:** 2025-01-29

**Authors:** Youn Young Shim, Peta-Gaye G. Burnett, Clara M. Olivia, Xian-Guo Zou, Sung Jin Lee, Hye-Jin Kim, Young Jun Kim, Martin J. T. Reaney

**Affiliations:** 1Department of Food and Bioproduct Sciences, University of Saskatchewan, Saskatoon, SK S7N 5A8, Canada; younyoung.shim@usask.ca (Y.Y.S.); pgb970@mail.usask.ca (P.-G.G.B.); cmo539@mail.usask.ca (C.M.O.); 2Prairie Tide Diversified Inc., Saskatoon, SK S7J 0R1, Canada; 3Department of Food and Biotechnology, Korea University, Sejong 30019, Republic of Korea; 4College of Food Science and Technology, Zhejiang University of Technology, Huzhou 313299, China; xianguozou2019@zjut.edu.cn; 5Food Science R & D Center, Kolmar BNH Co., Ltd., Seoul 06800, Republic of Korea

**Keywords:** oxidative stability, flaxseed, *Linum usitatissimum* L., linusorb, gluten-free, peptides, antioxidant

## Abstract

Flaxseed meal, rich in water-soluble gums, improves the texture of gluten-free (GF) products. Bioactive antioxidant peptides from flaxseed, known as linusorbs (LOs) or cyclolinopeptides, may provide health benefits. However, the stability of flaxseed-derived LOs during dough preparation, baking, and storage remains unclear. To investigate this, GF bread dough and bread were prepared with flaxseed meal, and the LO content was determined in the flaxseed meal, the bread flour with the flaxseed meal, the dough, and the bread. The LO levels were also monitored during storage at various temperatures (−18 °C, 4 °C, and 22–23 °C) for 0, 1, 2, and 4 weeks using high-performance liquid chromatography–diode array detection (HPLC-DAD). The levels of oxidized LOs, such as [1–9-NαC],[1-(*R*_s_,*S*_s_)-MetO]-linusorb B2 (LO**14**), remained relatively stable in the flaxseed meal and the flour derived from it across under all conditions for up to 4 weeks. Due to microbial contamination, the dough could not be stored at either 4 or 21 °C, and the bread could only be stored at 21 °C for one week. However, the bread and dough could be stored for up to 4 weeks at −18 °C, and the bread at 4 °C, without a significant loss of LOs. The main changes in LOs occurred during processing rather than storage. Reduced LOs were found in higher concentrations in the flour and meal compared to the dough and bread, without a corresponding increase in oxidized LOs. The flaxseed meal-fortified bread maintained oxidative stability when stored at low temperatures. This is the first study to investigate the effect of baking conditions on LO content and antioxidant properties.

## 1. Introduction

Celiac disease (CD) is a chronic inflammation of the small intestine caused by the ingestion of gluten [1], resulting in overt maldigestion and the malabsorption of nutritional components, such as minerals and fat-soluble vitamins [2], and even inducing extra-intestinal manifestations such as anemia, the reduction of bone mineral density, coagulopathy, arthralgia, and neurological disorders [3]. Its prevalence has been reported to occur in 1–2% of the general population [4]. Currently, the only effective treatment for CD is the strict and lifelong elimination of foods and medications containing gluten protein from wheat, barley, rye, and their derivatives [5].

Bread is a staple food consumed by much of the world’s population. Traditionally, bread is prepared from wheat flour that provides the gluten protein responsible for inducing the desirable visco-elastic properties [6]. A broad variety of gluten-free (GF) products made from rice flour, millet, tapioca flour, potato starch, and bean flour have been developed that can be consumed by individuals with CD [7]. Unfortunately, such GF flour products do not have the same cooking properties as products containing gluten, and do not have the same mouthfeel and taste as products made with gluten [8]. Interestingly, flaxseed has considerable amounts of soluble polysaccharides and protein [9]. The combined presence of flaxseed protein and gum forms a natural coacervate that can impart textural, sensory, and rheological properties to GF products [10]. Flaxseed gum can also interact with other food proteins [11] and improve the nutritional profile of GF foods [12], making it suitable as an additive in GF foods.

Flax (*Linum usitatissimum* L.) is an important oilseed crop that has been cultivated for food and feed purposes for over 5000 years. The chemical components of flaxseed, including α-linolenic acid (ALA), secoisolariciresinol diglucoside (SDG), phenolic compounds, proteins, and dietary fiber, all contribute to the health benefits arising from flax consumption [13]. The hydrophobic cyclopeptides of flaxseed, linusorbs (LOs), also have remarkable bioactivity. The reported LOs have eight to ten amino acids that are linked through an N- to C peptide bond and lack cysteine double bonds [14]. Studies show that LOs slow the oxidation of flaxseed oil, implying that they act as antioxidants [15,16]. Kaneda et al. [17] reported that six LOs suppressed osteoclast differentiation. Our recent studies revealed that [1–9-NαC],[1-MetO_2_]-linusorb B2 (LO**15**), [1–9-NαC]-linusorb B3 (LO**17**), and a mixture of LOs derived from flaxseed exhibited anti-tumor effects against SGC-7901 cells by triggering cell apoptosis and blocking cell cycles in the G1 phase through different signaling pathways [18,19,20]. In addition, the LOs induced anti-inflammatory activity in in vitro and in vivo assays through the downregulation of the NF-κB signaling pathway [21,22].

The physical properties and nutritional profile of flaxseed-enriched bakery and cereal-based products can be influenced during processing and storage. Conforti and Davis (2006) [23] found that the peroxide value of white and flaxseed-fortified bread increased to 1.53 and 2.92 meq. O_2_/kg, respectively, after storage at room temperature (RT) for 8 weeks. This finding indicates the oxidation of fatty acids in the bread. Lee et al. [24] reported that the content of conjugated dienes was lower in the dough enriched with ground flaxseed (0.29% of lipids) than that in the flaxseed-free dough (0.93% of the lipids), suggesting that antioxidants in flaxseed may prevent lipid oxidation during bread processing. Hall III et al. [25] investigated the changes in the ALA and SDG of flaxseed enriched macaroni during storage at RT and drying at higher temperatures (40, 70, and 90 °C), and found no differences in either the ALA or SDG content compared to the fresh product.

To design GF bakery products with desirable sensory properties and enhanced nutritional profiles, we prepared flaxseed meal-fortified GF bread as a model to study lignans and cyanogenic glycosides (CGs) [26]. Although CG decreased in the GF bread in our group’s preceding study, SDG, a phytoestrogen precursor lignan, was well retained in the flour, dough, and bread, even after storage for 4 weeks at temperatures between −18 °C and 22–23 °C [26]. However, it is unclear whether the LOs in flaxseed-fortified GF bread are oxidatively stable. Therefore, this study is designed to determine the fate of LOs in flaxseed-fortified GF products during bread production and storage at different temperatures as described previously [26]. The results from this study will help to determine the potential of using flaxseed meals in GF bakery food products with known nutraceuticals.

## 2. Materials and Methods

### 2.1. Materials

Acetonitrile (CH_3_CN) and methanol (MeOH) of chromatographic grade were purchased from Fisher Scientific International Inc. (Fair Lawn, NJ, USA). Flaxseed meal (*L. usitatissimum* L. cv CDC Sorrel) and the purified LOs (Figure 1; Table 1) were kindly provided by Prairie Tide Diversified Inc. (Saskatoon, SK, Canada). Deionized reversed osmosis (RO) water (resistivity > 18.2 MΩ∙cm at 25 °C) was prepared by a Milli-Q RO system from Millipore (Bedford, MA, USA). Solvents of analytical reagent grade were obtained from Sigma-Aldrich (St. Louis, MO, USA).

### 2.2. Sample Preparation and Collection for Chemical Analysis

Table 2 lists the GF bread ingredients for the bread samples prepared with and without flaxseed meal [28]. The control GF bread without flaxseed meal, was prepared by thoroughly mixing white rice flour, potato starch, and tapioca flour with sugar, yeast, and salt at medium speed using a Kitchen Aid Ultra Power Mixer (Kitchen Aid, St. Joseph’s, MI, USA). Warm milk (45 °C) and butter were then added to this mixture and blended at a low speed. The dough was formed by adding the eggs one by one and mixing at a high speed for 3 min. The thoroughly mixed dough was poured into greased loaf pans, which were then covered and kept at 22–23 °C to rise to double the volume. The leavened dough was baked in an oven at 175 °C for 40 min. After baking, the GF bread was immediately removed from the loaf pan and allowed to cool at RT for 10 min. To prepare the flaxseed meal-fortified GF bread, the same procedure was followed, except that 20% (*w*/*w*) of the flour mixture (rice flour, potato starch, and tapioca flour) was replaced with flaxseed meal. Samples of the GF flour, dough, and bread were taken at each stage of the preparation process, both with and without the substitution of rice flour with flaxseed meal and analyzed on the day of preparation (0 d). After preparation, all test samples (flaxseed meal and fortified flour, dough without yeast, and GF bread) were placed in plastic bags (Ziploc, Racine, WI, USA), and stored at RT (22–23 °C), in a refrigerator (4 °C), and in a freezer (−18 °C). An analysis of the test samples was conducted either immediately on the day of preparation (0 d), or after storage for 1, 2, and 4 weeks.

### 2.3. Extraction of LOs

All test bread samples (2.0 g) were pre-ground using a coffee grinder (VWR, Radnor, PA, USA) before weighing. Anhydrous MeOH (6.5 mL) was added to the wet dough samples to achieve a dilution of approximately 70% MeOH. Aqueous 70% MeOH (10 mL, *v*/*v*) was added to the other samples. The internal standard LO**16** (300 µL, 0.1 mg/mL) was separately added to the mixture. The LOs were then extracted at 60 °C for 2 h in a water bath (VWR model 1350 GM, Cornelius, OR, USA). After extraction, the samples were cooled at 22–23 °C for 30 min and then centrifuged at 3000× *g* for 10 min (Beckman-Coulter Allegra X-22R, Palo Alto, CA, USA).

### 2.4. High-Performance Liquid Chromatography (HPLC) Analysis of LOs

The separation and analysis of LOs were performed on an Agilent 1200 series HPLC system (Agilent Technologies Canada Inc., Mississauga, ON, Canada). The system included a quaternary pump, an autosampler, a Chromolith^®^ SpeedROD RP-C_18_ column (100 mm × 4.6 mm, i.d., Merck KGaA, Darmstadt, Germany), a photodiode-array detector, and a degasser. The mobile phase consisted of deionized water (A) and acetonitrile (B), with the following gradient: 0 min, 30% B; 4.0 min, 70% B; 4.5 min, 90% B; 5.0 min, 30% B; and 6.0 min, 30% B. The flow rate was 2.0 mL/min, and the column temperature was maintained at 32 °C. Eluting peaks were monitored at 214 nm (10 nm bandwidth) and 244 nm (20 nm bandwidth), with a reference signal at 360 nm (100 nm bandwidth). Data analysis was performed using ChemStation LC 3D system software (G1701DA D.01.00, Agilent Technologies Inc., Mississauga, ON, Canada).

### 2.5. Statistical Analysis

Statistical analyses were performed using the Statistical Package for the Social Sciences (SPSS) software version 18.0 (SPSS Inc., Chicago, IL, USA). Data were presented as mean ± standard deviation (SD), with *n* = 3. The differences between means were assessed using a one-way analysis of variance (ANOVA), followed by a post-hoc least significant difference test, or an unpaired Student’s *t*-test. Statistical significance was set at *p* < 0.05.

## 3. Results and Discussion

### 3.1. Analysis of LO in Flaxseed-Fortified Bread

When consumed, the nutraceutical compounds present in flaxseed can help reduce the incidence of cardiovascular and gastrointestinal tract diseases, osteoporosis, and diabetes, and regulate blood serum cholesterol [13,29,30]. Bakery and cereal-based products, such as bread, macaroni, and pasta, are good representative candidates for flaxseed enrichment as it is readily added to formulations without producing an obvious negative impact on food quality [31,32]. However, challenges in the development of flaxseed-fortified bakery and cereal products include the preservation of health-promoting flaxseed constituents during processing and storage. Numerous reports indicated that bakery and cereal products fortified with flaxseed generally exhibit similar or even an improved shelf life compared to the corresponding products without fortification. Similarly, nutritional components, such as SDG and lipids, generally remain stable through a range of processes and various storage conditions [10,25,33,34]. LOs are bioactive compounds found in flaxseed that have the potential to treat inflammation, cancer, and other diseases and modulate immune responses [35]. These compounds are also antioxidants. Methionine (Met)-containing LOs can be oxidized to analogs bearing a Met sulfoxide (*S*-oxide), MetO. When oxidation conditions are more severe (62–65 °C and 100 °C), the Met in LOs can be oxidized to Met sulfone (*S*,*S*-oxide), MetO_2_ [36,37,38]. Aladedunye et al. [36] found no significant changes in the total and individual LOs of flaxseed meal stored at ambient temperatures for up to 48 months. Nevertheless, it is unknown how the LOs changed in the GF bread flour, dough, and bread fortified with flaxseed (flour, dough, and bread) and under the different storage conditions (temperature and time).

### 3.2. HPLC Chromatogram of Flaxseed-Fortified Flour and the Control

The separation and identification of LOs from flaxseed meal extracts is possible using conventional reversed-phase HPLC columns; however, such columns are not suitable or convenient for a high-throughput analysis [39,40]. Olivia et al. [41] compared the separation of LOs from a mixture using HPLC columns with different properties (a conventional reversed-phase HPLC column with 5 micrometer beads, a reversed-phase monolithic HPLC column, and a perfusion column) and found that the monolithic columns (Chromolith^®^ SpeedROD) provided a better resolution of the LOs while achieving a longer column life than possible with packed columns. Monolithic columns were deemed more suitable than packed columns for high throughput LO screening. In this study, flaxseed meal and the samples collected during processing, and storage of flaxseed-fortified GF bread were extracted with an aqueous methanolic solution (70%, *v*/*v*) at a temperature of 60 °C. The LOs in the extract were separated using a Chromolith^®^ SpeedROD column. The HPLC chromatograms of the flaxseed-fortified flour and control flour indicate that there are no LOs in the control flour. However, LO peaks **1**–**17** at the retention times from 2.5 to 5.0 min in the flaxseed-fortified flour were observed (Figure 2).

The internal standard [1–9-NαC],[1-Abu]-linusorb B2 (LO**16**) was a partially synthetic LO included to minimize interference from the sample matrix during preparation for the analysis [35,41]. The internal standard **16** lacked the less stable amino acids, Met and tryptophan (Trp), and might be seen as a close analog of **17**. The internal standard had relatively low polarity and was a superior standard for determining matrix effects on the LOs with a similar polarity. The analysis of variance of the internal standard recovery revealed that the variance of standard recovery from the dough was greater than from the other matrices. It is possible that the wetted dough affected the extraction. For each type of sample, the efficiency of the internal standard recovery was constant throughout the experiment, with all differences attributable to statistical variance (Table 3). The internal standard recovery from the dough samples from week 0 was higher than from the meal, flour, and bread. The recovery of the internal standard from the bread sample on week 4 was lower than on week 2.

It was not clear from the analyses if the Met or Trp present in **1**–**15** affected compound recovery or stability. The compounds with one Trp and two Met residues include the fully reduced species **1**, and **4**. In addition, compounds **9** and **3** with one Met and two MetO residues were resolved. In the region of the chromatogram where the partially oxidized forms of compounds **1** and **4** elute, there is a complex series of overlapping peaks that proved difficult to interpret. Other chromatographic media have successfully separated many of the partially oxidized LOs compounds [42].

#### 3.2.1. Expected Ratios of LOs

All of the observed LOs present in flaxseed meals are the products of ribosomal synthesis followed by post-translational modification. As such, the ratios of LOs are determined by the ratios of the LOs in the precursor genes [43]. Furthermore, ribosomal synthesis adds Met but not MetO to the peptides and thus the occurrence of any MetO can be seen as either a post-translational modification or oxidation. We, and others, have observed that fresh flaxseed products from new seed have very little MetO. Therefore, it is expected that the following ratios of LOs should be expected. In the cultivar CDC Sorrel, one precursor peptide is the source of compounds **1**, **4**, and **6** and the other precursor peptide encodes **10**, **13**, and **17**. Each precursor for compounds **1**, **4**, and **6** produces six LOs with three being **1**, two being **4** and one being **6**. These ratios are clearly present in the observed products. Similarly, the ratio of LOs **10**, **13**, and **17** in the precursor is 1:1:1 in CDC Sorrel.

#### 3.2.2. Pooling Data

For the flaxseed meal and flour stored at three temperatures over 4 weeks, no statistically significant changes were observed in the concentration of any LO peak. All the analyses could be interpreted as a repeated analysis of the same sample. When ground flaxseed meal was blended with flour, and the flour held for 4 weeks, the same finding of no change in the LOs over the 4 weeks period, and three storage temperatures was made. However, the areas of all the peaks observed in the chromatograms were considerably lower for the flaxseed flour-fortified meal than for the flaxseed meal. The lower areas of all the peaks were attributed to the dilution of the flaxseed meal by the other flour components. The flaxseed meal was 28% of the flour mass. A two-factor analysis of variance was conducted on the pooled data with *n* = 60 to compare the individual LO levels and the dilution caused by blending the flaxseed meal with other ingredients to make flour. Based on observations of the chromatography of the control flour extracts with no added flaxseed meal, interfering peaks contributed by the flour made the observation of the peak areas contributed by **5**, **8**, and **9** unreliable in the samples with flour added.

The production of dough from the flour creates a product that is further diluted. As the dough was a perishable product, it was analyzed immediately or frozen and analyzed at 1, 2, and 4 weeks. There were no differences among the LOs measured in the fresh and frozen dough samples. Therefore, for a comparison of the dough to meal-fortified flour and bread, all the data were pooled and treated as replicates.

The dough is prepared by blending dry flour ingredients with two additional dry ingredients (yeast and salt) and three moisture-bearing ingredients (butter, eggs, and milk). The additional dry and moist ingredients further dilute the LOs to 71% of the concentration present in flour. The LOs in the dough were measured to be 61% of the LOs in the flour. The most reduced LOs, those with the lowest polarity (**1**, **4**, **6**, **10**, **13**, and **17**), decreased most compared to the concentration recovered from the flour. The ratio of the peak areas representing these low polarity LOs ranged from 0.318 to 0.377. In comparison, the concentrations of more oxidized and consequently polar LOs in the dough were higher in the extracts compared to the less polar LOs. The ratios of the polar LO peak areas in the dough compared to the flour were from 0.511 to 0.941. This range in numbers might be attributed to the oxidation of the less polar species to the more polar species contributing each peak as well as the matrix effects that are possibly different between the flour and the dough. All the ratios were below one, and this reflects the expected dilution of the starting flour by the added ingredients. The bread was then made from the flaxseed-fortified dough. Typically, bread loses moisture during cooking, and some concentration of LOs is possible with the loss in mass. Conversely, with the exposure to heat and oxidants during cooking, a portion of the LOs might be degraded. It was, therefore, interesting that the LO concentration present in the bread was only slightly lower than the concentration in the dough. Furthermore, there was no evidence that the LOs were oxidized during baking (Table 4 and Table 5).

### 3.3. Effects of Storage Temperature and Time

To account for the possible matrix effects contributed by each matrix, all HPLC peak areas were divided by the area of **16.** Therefore, all data are presented as unitless numbers. As was expected, the concentration of LOs in the flaxseed meal samples held for 4 weeks at −18 °C, 4 °C, and 22–23 °C were not affected by the treatments. The predominant peaks in these samples were contributed by **1**, **4**, **6**, **10**, **13**, and **17**. Under the conditions used for chromatography, compounds **10** and **17** were not resolved and compounds **1** and **6** were also not resolved. Therefore, the four peaks represent all the compounds present in their reduced state. Compounds **10** and **13** have a Met group but no Trp and can occur as a singly oxidized MetO (**11** and **14**) or doubly oxidized to MetO_2_ (**12** and **15**). Although not shown here, there was no evidence of the presence of MetO_2_ compounds. The MetO compounds **11** and **14** are well resolved from possible interfering peaks.

## 4. Conclusions

Flaxseed is a rich source of a portfolio of bioactive compounds that are important in a diet for maintaining health. In this study, flaxseed meal was incorporated into GF bakery bread, and the concentration of bioactive LOs at each stage of processing and at each time interval was determined. Firstly, LOs were detected in all the flaxseed meal-fortified samples during processing and storage. The LO concentrations in the flaxseed meal-containing products, with flour containing 28% dry weight (dw) flaxseed meal, dough containing 20% (dw) flaxseed meal, and bread containing over 20% (dw) flaxseed meal, were all below the expected dilutions. While dilution has a measurable effect on the content of all LOs, there was no effect noted for the storage time in the flour or flaxseed meal over the times and temperatures tested. As both the dough and bread were perishable, the persistence of LOs was not determined in these products, but the LO content was oxidatively stable in the frozen dough and bread. The bread refrigerated for one week also had a similar LO content to the fresh or frozen bread. The readily observable LOs present in the flaxseed meals and flour were highly correlated, indicating that blending the dry ingredients did not affect the LOs. However, the addition of wet ingredients to produce dough and bread profoundly changed the profile of the LOs as observed. The concentrations of low polarity reduced LOs were significantly lower in the extracts of the dough and bread compared to the expected levels from the simple dilution of the flaxseed meal LOs. The lower levels of LOs observed in the bread and dough might have been caused by poor extraction, oxidation, or by a high affinity for or irreversible binding to the dough and bread matrices. We believe that poor extraction is unlikely as the internal standard has a similarly low polarity to the LOs in low abundance. Further, we believe that simple oxidation is also unlikely as the highly oxidized, more polar species in the bread and dough appeared at concentrations close to the expected dilution. Thus, it appears that it is most likely that the reduced LO concentration is related to some form of affinity of the LOs to the dough and bread matrices. Regardless of the lower level of LOs observed in the bread compared to the flaxseed meals, flaxseed LOs persist throughout the process of making bread and dough. However, this study was limited to the analysis of flaxseed meal-fortified GF bread under specific conditions. Further research is needed to investigate the long-term stability of the bioactive compounds in flaxseed under various storage and processing conditions. Additionally, the impact of flaxseed incorporation on the sensory properties and consumer acceptance of GF bakery products warrants further exploration. This study suggests for the first time that flaxseed bread is a sufficient dietary source of cyclic peptides.

## Figures and Tables

**Figure 1 foods-14-00439-f001:**
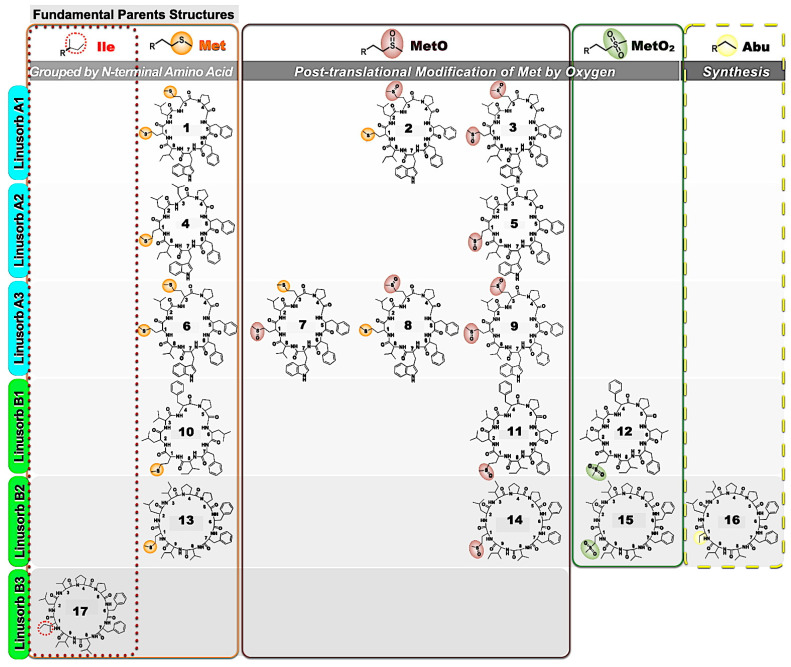
Chemical structures of LOs (**1**–**17**).

**Figure 2 foods-14-00439-f002:**
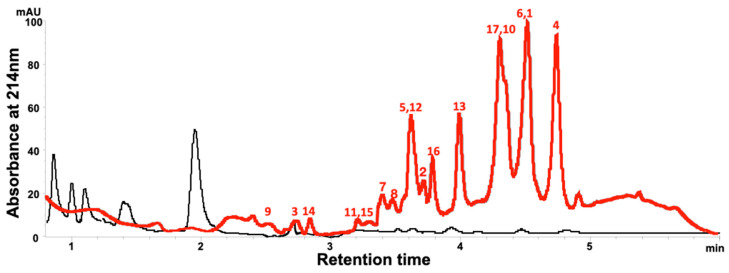
HPLC-DAD chromatogram of control flour (black) and a flaxseed meal-fortified flour stored for 4 weeks in a refrigerator (red). Identification of LO peaks **1**–**17** is based on previous studies [32,38].

**Table 1 foods-14-00439-t001:** LO nomenclature used.

LO Name *^a^*	Literature Name *^b^* (Code)	Amino Acid Sequence (NαC-) *^c^*
[1–8-NαC]-linusorb A1	CLM (**1**)	Met-Leu-Met-Pro-Phe-Phe-Trp-Ile
[1–8-NαC],[3-(*R*_s_,*S*_s_)-MetO]-linusorb A1	CLN (**2**)	Met-Leu-[(*R*_s_,*S*_s_)-MetO]-Pro-Phe-Phe-Trp-Ile
[1–8-NαC],[1,3-(*R*_s_,*S*_s_)-MetO]-linusorb A1	CLG (**3**) *^d^*	[(*R*_s_,*S*_s_)-MetO]-Leu-[(*R*_s_,*S*_s_)-MetO]-Pro-Phe-Phe-Trp-Ile
[1–8-NαC]-linusorb A2	CLO (**4**)	Met-Leu-Leu-Pro-Phe-Phe-Trp-Ile
[1–8-NαC],[1-(*R*_s_,*S*_s_)-MetO]-linusorb A2	CLD (**5**) *^e^*	[(*R*_s_,*S*_s_)-MetO]-Leu-Leu-Pro-Phe-Phe-Trp-Ile
[1–8-NαC]-linusorb A3	CLL (**6**)	Met-Leu-Met-Pro-Phe-Phe-Trp-Val
[1–8-NαC],[1-(*R*_s_,*S*_s_)-MetO]-linusorb A3	[1–8-NαC],[1-MetO]-CLF (**7**)	[(*R*_s_,*S*_s_)-MetO]-Leu-Met-Pro-Phe-Phe-Trp-Val
[1–8-NαC],[3-(*R*_s_,*S*_s_)-MetO]-linusorb A3	CLI (**8**)	Met-Leu-[(*R*_s_,*S*_s_)-MetO]-Pro-Phe-Phe-Trp-Val
[1–8-NαC],[1,3-(*R*_s_,*S*_s_)-MetO]-linusorb A3	CLF (**9**) *^d^*	[(*R*_s_,*S*_s_)-MetO]-Leu-[(*R*_s_,*S*_s_)-MetO]-Pro-Phe-Phe-Trp-Val
[1–8-NαC]-linusorb B1	CLP (**10**) *^f^*	Met-Leu-Val-Phe-Pro-Leu-Phe-Ile
[1–8-NαC],[1-(*R*_s_,*S*_s_)-MetO]-linusorb B1	CLE (**11**)	[(*R*_s_,*S*_s_)-MetO]-Leu-Val-Phe-Pro-Leu-Phe-Ile
[1–8-NαC],[1-MetO_2_]-linusorb B1	CLJ (**12**)	MetO_2_-Leu-Val-Phe-Pro-Leu-Phe-Ile
[1–9-NαC]-linusorb B2	CLB (**13**) *^f^*	Met-Leu-Ile-Pro-Pro-Phe-Phe-Val-Ile
[1–9-NαC],[1-(*R*_s_,*S*_s_)-MetO]-linusorb B2	CLC (**14**)	[(*R*_s_,*S*_s_)-MetO]-Leu-Ile-Pro-Pro-Phe-Phe-Val-Ile
[1–9-NαC],[1-MetO_2_]-linusorb B2	CLK (**15**)	MetO_2_-Leu-Ile-Pro-Pro-Phe-Phe-Val-Ile
[1–9-NαC],[1-Abu]-linusorb B2	CLB-S (**16**)	Abu-Leu-Ile-Pro-Pro-Phe-Phe-Val-Ile
[1–9-NαC]-linusorb B3	CLA (**17**) *^g^*	Ile-Leu-Val-Pro-Pro-Phe-Phe-Leu-Ile

*^a^* The LO name is formatted as [1–#-NαC], indicating a linkage between amino acid 1 and amino acid “#” through the α-amino group, resulting in an N-C cyclization of the core peptide. Use the en dash (–) for ranges and place them in square brackets: [#-Xaa,#-Xaa] (green). Methionine *S*-oxide contains a chiral sulfur with two diastereomers designated as *S*_s_ and *R*_s_. The identical amino acid substituents are numbered and grouped. The genus name is abbreviated to three letters or from the UniProt list (lin-), and the species name is recognized by two letters or the name from the UniProt list (-us-). The common suffix for orbitides is -orb. *^b^* The name used in the first literature description is indicated. *^c^* Methionine residues in the amino acid sequences are highlighted in Figure 1. *^d^* LO**3** and LO**9** contain one Trp and two MetO; *^e^* LO**5** has one Trp and one MetO; *^f^* LO**10** and LO**13** have no Trp and one Met; *^g^* LO**17** has no Trp and no Met. Abbreviations for the methionine residues are MetO for methionine *S*-oxide and MetO_2_ for methionine *S*,*S*-dioxide. The information is modified from Shim et al. [27].

**Table 2 foods-14-00439-t002:** Components of GF-bread and flaxseed meal-fortified GF-bread.

Ingredient	GF-Bread (Control, g)	Flaxseed Meal-Fortified GF-Bread (g)
White rice flour	846	1054
Potato starch	194	389
Tapioca flour	82	164
Flaxseed meal	0	637
Sugar	12	24
Total dry ingredients	1134	2268
Instant yeast	6	13
Salt	7	14
Milk (45 °C)	242	484
Unsalted butter	57	113
Eggs	146	293
Total wet ingredients	458	917
Total ingredients	1592	3185

**Table 3 foods-14-00439-t003:** ANOVA of the internal standard (LO**16**).

Samples		Meal	Flour	Dough	Bread	Total
Week 0	Average	115	125	161	109	127
	Variance	0.1	32	34	25	471
Week 1	Average	135	102	141	141	130
	Variance	334	44	512	419	530
Week 2	Average	116	123	162	128	132
	Variance	527	34	1112	23	646
Week 4	Average	112	100	105	88	101
	Variance	0.2	28	243	3	132
Total	Average	119	113	142	116	NA
	Variance	258	172	927	524	NA

NA: Not available.

**Table 4 foods-14-00439-t004:** Area ratios of peaks (mean ± SD) contributed by specific LOs compared to the internal standard (**16**).

LO	Area Meal	Area Flour	Area Dough	Area Bread
**9**	1.175 ± 0.004	0.280 ± 0.001	0.127 ± 0.000	0.246 ± 0.001
**3**	3.063 ± 0.422	0.844 ± 0.090	1.011 ± 0.014	0.564 ± 0.000
**14**	1.751 ± 0.124	0.462 ± 0.022	0.176 ± 0.000	0.283 ± 0.000
**11, 15**	1.693 ± 0.156	0.447 ± 0.007	0.320 ± 0.006	0.215 ± 0.001
**7**	2.403 ± 0.222	0.524 ± 0.005	0.357 ± 0.003	1.383 ± 0.061
**8**	1.062 ± 0.101	0.487 ± 0.002	0.312 ± 0.000	0.652 ± 0.004
**5, 12**	5.983 ± 0.282	2.643 ± 0.112	1.591 ± 0.010	1.401 ± 0.005
**2**	2.414 ± 0.076	0.581 ± 0.002	0.407 ± 0.002	0.490 ± 0.002
**16**	1.000 ± 0.000	1.000 ± 0.000	1.000 ± 0.000	1.000 ± 0.000
**13**	5.663 ± 2.262	1.647 ± 0.048	0.631 ± 0.001	0.677 ± 0.003
**17, 10**	19.44 ± 1.924	4.783 ± 0.362	1.850 ± 0.005	1.714 ± 0.004
**6, 1**	18.12 ± 1.502	4.141 ± 0.360	1.356 ± 0.005	1.141 ± 0.009
**4**	14.12 ± 1.122	3.173 ± 0.134	1.334 ± 0.004	1.119 ± 0.001

**Table 5 foods-14-00439-t005:** Area of the ratios of peaks comparing the LOs in flaxseed meal (mean ± SD), flaxseed flour, flaxseed dough, and flaxseed bread.

LO	Area Flour/Area Meal	Area Dough/Area Flour	Area Bread/Area Dough
**9**	0.231 ± 0.001	0.941 ± 0.051	0.936 ± 0.061
**3**	0.267 ± 0.002	0.866 ± 0.077	0.776 ± 0.005
**14**	0.256 ± 0.001	0.511 ± 0.002	1.206 ± 0.021
**11, 15**	0.257 ± 0.001	0.744 ± 0.006	0.681 ± 0.003
**7**	0.215 ± 0.002	0.661 ± 0.001	1.014 ± 0.011
**8**	0.453 ± 0.003	0.675 ± 0.001	0.777 ± 0.006
**5, 12**	0.441 ± 0.003	0.596 ± 0.004	0.891 ± 0.081
**2**	0.242 ± 0.001	0.896 ± 0.071	0.945 ± 0.009
**13**	0.271 ± 0.001	0.376 ± 0.002	1.101 ± 0.004
**17, 10**	0.243 ± 0.001	0.377 ± 0.001	0.952 ± 0.011
**6, 1**	0.226 ± 0.001	0.318 ± 0.003	0.903 ± 0.018

## Data Availability

The original contributions presented in this study are included in the article. Further inquiries can be directed to the corresponding authors.

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
