# Peer review of "Oxidative Stability of Novel Peptides (Linusorbs) in Flaxseed Meal-Fortified Gluten-Free Bread"

_foods, 2025, doi:10.3390/foods14030439_

Round 1

Reviewer 1 Report

Comments and Suggestions for Authors

Introduction

Overall, the introduction is well-prepared. It has a logical, concise structure.

The reviewer suggests changes in lines 82-87, as the fragment does not sound good.

Additionally, what does the term "safe" mean in line 82?

Materials and Methods:

This section is generally well described.

Please explain why some ingredients are in higher amounts in the flaxseed-enriched version.

Statistical methods – according to the reviewer, it would be helpful to indicate what variables were analyzed.

Results and Discussion: Well described, the text raises no doubts.

Conlusions

The conclusions appear to be fundamentally correct, as they are based on the presented results and align with the data regarding the stability of bioactive compounds in flaxseed under various conditions and the impact of production processes on their content, but conclusions could be expanded to include a broader practical context and added the study's limitations. In their current form, they summarize the results accurately but must fully address how to utilize flaxseed in food products best to preserve its full health benefits.

Author Response

Reviewer #1:

Comment: Introduction, overall, the introduction is well-prepared. It has a logical, concise structure. The reviewer suggests changes in lines 82-87, as the fragment does not sound good.

Response: We revised lines 81-82 toTo design GF bakery products with desirable sensory properties and enhanced nutritional profiles, we prepared flaxseed meal-fortified GF bread as a model to study of lignans and cyanogenic glycosides (CGs)”

Comment: Additionally, what does the term "safe" mean in line 82?

Response We removed the word safe. Note that this referred to our observation that CGs were depleted in bread making.

Comment: Materials and Methods, this section is generally well described. Please explain why some ingredients are in higher amounts in the flaxseed-enriched version.

Response We prepared twice as much flaxseed meal-fortified gluten-free (GF) bread compared to the control version. This resulted in increased amounts of all ingredients to maintain consistency in the preparation process. Additionally, the inclusion of flaxseed meal displaced white rice flour in the recipe, which contributed to the observed changes in ingredient quantities.

 Comment: Materials and Methods, statistical methods – according to the reviewer, it would be helpful to indicate what variables were analyzed.

Response The expected LOs that can be identified (See Table 1) are reported in Table 4.

Comment: Results and Discussion: well described, the text raises no doubts.

Response: Thank you

Comment: Conclusions: The conclusions appear to be fundamentally correct, as they are based on the presented results and align with the data regarding the stability of bioactive compounds in flaxseed under various conditions and the impact of production processes on their content, but conclusions could be expanded to include a broader practical context and added the study's limitations. In their current form, they summarize the results accurately but must fully address how to utilize flaxseed in food products best to preserve its full health benefits.

Response: We added the Study Limitations (Lines 320-324). “This study was limited to the analysis of flaxseed meal-fortified GF bread under specific conditions. Further research is needed to investigate the long-term stability of bioactive compounds in flaxseed under various storage and processing conditions. Additionally, the impact of flaxseed incorporation on the sensory properties and consumer acceptance of GF bakery products warrants further exploration.”

Reviewer 2 Report

Comments and Suggestions for Authors

Your work is interesting, but I think your presentation of your results does not demonstrate it. You could enrich your work with additional analyses that provide more information, perhaps even more. It would be interesting to report the content of LOs in concentration rather than as areas. 

The references could be updated since only seven of the 42 references are from 2019  to date. 

It would be interesting to place the chromatograms where the changes in the formation of LOs can be appreciated.

Author Response

Reviewer #2:

Comment: Your work is interesting, but I think your presentation of your results does not demonstrate it. You could enrich your work with additional analyses that provide more information, perhaps even more. It would be interesting to report the content of LOs in concentration rather than as areas. 

Response: We are reluctant to publish exact concentrations as this would require standards for the quantification of each compound, which were not available for this study. However, in our previous research [43], we have demonstrated that the molar absorbance at 214 nm best reflects the proportional concentration of the peptide LOs 1–17. Therefore, we have reported the content of LOs as areas, which provides a reliable and proportional representation of their concentrations.

[43] Gui, B.; Shim, Y.Y.; Datla, R.S.S.; Covello, P.S.; Stone, S.L.; Reaney, M.J.T. Identification and quantification of cyclolinopeptides in five flaxseed cultivars. J. Agric. Food Chem. 201260 (35), 8571–8579. https://doi.org/10.1021/jf301847u

Comment: The references could be updated since only seven of the 42 references are from 2019 to date. 

Response: We have updated the references [2–4, 43].

[2] Lebwohl, B.; Sanders, D.S.; Green, P.H.R. Coeliac disease. Lancet 2018391 (10115), 70–81. https://doi.org/10.1016/S0140-6736(17)31796-8.

[3]  Caio, G.; Volta, U.; Sapone, A.; Leffler, D.A.; De Giorgio, R.; Catassi, C.; Fasano, A. Celiac disease: a comprehensive current review. BMC Med. 201917, 1–20. https://doi.org/10.1186/s12916-019-1380-z.

[4]  Singh, P.; Arora, A.; Strand, T.A.; Leffler, D.A.; Catassi, C.; Green, P.H.; Kelly, C.P.; Ahuja, V.; Makharia, G.K. Global prevalence of celiac disease: systematic review and meta-analysis. Clin. Gastroenterol. Hepatol. 201816 (6), 823–836. https://doi.org/10.1016/j.cgh.2017.06.037.

[43] Gui, B.; Shim, Y.Y.; Datla, R.S.S.; Covello, P.S.; Stone, S.L.; Reaney, M.J.T. Identification and quantification of cyclolinopeptides in five flaxseed cultivars. J. Agric. Food Chem. 201260 (35), 8571–8579. https://doi.org/10.1021/jf301847u

Comment: It would be interesting to place the chromatograms where the changes in the formation of LOs can be appreciated. 

Response: We are not certain what is meant by this statement.

Reviewer 3 Report

Comments and Suggestions for Authors

General comments:  The manuscript describes the changes that occurred in flaxseed peptides when flaxseed meal is used as an ingredient in bread doughs.  Degradation of the peptides was found to occur due to the heat of baking and that storage did not influence them.  No increase in storage life was found.  It was difficult to understand what the authors determined about the stability of the LO.  The question of the suitability of the extraction for the analysis makes any interpretation highly speculative.

Table 1:  Title is awkward.  Suggest “Table 1. Linusorbs (LO) used

Figure 1:  Is there a reference for this figure?  Was it created by the authors?

Section 3.2:  Why were no authentic standards of the LO run with the samples to confirm identifications?

Section 3.2.1:  What is the reference for this?

Lines 316-317:  If the LO had affinity to the matrices, would that mean that the extraction methodology was not suitable for this study?

References:  Check format instructions to the authors.  Not consistent in what is being used.

Line 353:  Change to “J. Amer. Diet. Assoc.

Author Response

Reviewer #3:

Comment: General comments: The manuscript describes the changes that occurred in flaxseed peptides when flaxseed meal is used as an ingredient in bread doughs. Degradation of the peptides was found to occur due to the heat of baking and that storage did not influence them. 

Response: We would like to address your comment regarding the degradation of peptides due to the heat of baking. Based on Table 5 of our manuscript, the ratio of peptides in dough and bread was very similar, indicating that the heat of baking did not significantly degrade the peptides. Our findings suggest that the concentration of LOs in the bread was only slightly lower than in the dough, and there was no evidence of oxidation during baking, as detailed in Tables 4 and 5. We believe this data supports our conclusion that the heat of baking did not substantially influence the degradation of peptides. We hope this clarifies our findings and addresses your concerns.

Comment: No increase in storage life was found. It was difficult to understand what the authors determined about the stability of the LO. The question of the suitability of the extraction for the analysis makes any interpretation highly speculative.

Response: We do not believe that we have speculated in our interpretation of results. We concluded that the bread and dough could not be stored at room temperature. For dry flour and meal there was no degradation with storage at any temperature. For frozen dough and bread, the results were similar, showing no degradation. We did note that dough and bread had a finite shelf life at room temperature. We did not analyze the perishable dough and bread products at higher temperatures.

Comment: Table 1: Title is awkward. Suggest “Table 1. Linusorbs (LO) used

Response: Changed accordingly (Line 100).

Comment: Figure 1: Is there a reference for this figure? Was it created by the authors?

Response: Figure 1 was created by the author, Youn Young Shim.

Comment: Section 3.2: Why were no authentic standards of the LO run with the samples to confirm identifications?

Response: We have reported previously regarding the elution order of standard compounds [41]. The elution of individual LOs on C-18 reversed phase is known as confirmed by MS analysis [37].

[37] Jadhav, P.D.; Okinyo-Owiti, D.P.; Ahiahonu, P. W.; Reaney, M.J.T. Detection, isolation and characterisation of cyclolinopeptides J and K in ageing flax. Food Chem. 2013, 138, 1757–1763. https://doi.org/10.1016/j.foodchem.2012.10.126.

[41] Olivia, C.M.; Burnett, P.-G.G.; Okinyo-Owiti, D.P.; Shen, J.; Reaney, M.J.T. Rapid reversed-phase liquid chromatography separation of cyclolinopeptides with monolithic and microparticulate columns. J. Chromatogr. B 2012, 904, 128–134. https://doi.org/10.1016/j.jchromb.2012.07.037.

Comment: Section 3.2.1: What is the reference for this?

Response: Cited the reference [43] and inserted it into the reference list (Line 229).

[43] Gui, B.; Shim, Y.Y.; Datla, R.S.S.; Covello, P.S.; Stone, S.L.; Reaney, M.J.T. Identification and quantification of cyclolinopeptides in five flaxseed cultivars. J. Agric. Food Chem. 201260 (35), 8571–8579. https://doi.org/10.1021/jf301847u

Comment: Lines 316-317: If the LO had affinity to the matrices, would that mean that the extraction methodology was not suitable for this study?

Response: Matrix effects are common in the extraction of compounds.

Comment: References: Check format instructions to the authors. Not consistent in what is being used.

Response: Revised.

Comment: Line 353: Change to “J. Amer. Diet. Assoc.

Response: The existing Ref for 2008 was replaced with the Ref for 2019.

[3] Caio, G.; Volta, U.; Sapone, A.; Leffler, D.A.; De Giorgio, R.; Catassi, C.; Fasano, A. Celiac disease: a comprehensive current review. BMC Med. 201917, 1–20. https://doi.org/10.1186/s12916-019-1380-z.

Round 2

Reviewer 3 Report

Comments and Suggestions for Authors

Changes made as requested.